# TABX: X-CELLENT AT COMPLEX TABLES AND BEYOND

## ABSTRACT

Recent advances in table understanding have shifted from text-based large language model (LLM) methods to multimodal LLM (MLLM) methods like Table-LLaVA that directly process table images. Despite these advances, existing table MLLMs still exhibit limited robustness to complex table layouts and poor generalization to unseen tasks. We trace these failings to two fundamental issues in their development pipeline: (1) a low-quality dataset composed of instruction-table-answer triplets and (2) a lack of all-around understanding of table images. This predicament is analogous to a student learning from flawed material with no mechanism for self-correction. Typically, true understanding is not attained through passive study alone, but rather through iterative self-evaluation and the correction of errors under teacher guidance. Inspired by this cognitive process, we first curate a new dataset, MMTab-Pro, by introducing three challenging tuning tasks that encourage the model to perform a deeper understanding of table content and structure, while applying a reflection-based enhancement to refine low-quality triplets. We further propose a Self-Evolution with Teacher-Tuning (SETT) framework to fine-tune the model, which enables the model to evolve through self-feedback and the guidance of a stronger teacher model, continuously refining both data suitability and model comprehension. Finally, through the two-step pipeline developed above, we present **TabX**, a robust and generalizable table MLLM. Experiments on the MMTab-eval benchmark show that **TabX** outperforms existing models, particularly on structurally complex and unseen tasks.

## 1 INTRODUCTION

Tables serve as an efficient means of organizing and storing data, widely used across various real-world scenarios such as finance, e-governance, and scientific research. They encapsulate complex and dense information in a structured format, forming a crucial basis for human knowledge acquisition and decision-making. With the growing volume of tabular data, diverse table understanding tasks have been actively explored, such as table-based question answering Cheng et al. (2022); Nan et al. (2022), text generation Parikh et al. (2020), and schema augmentation Zhang & Balog (2017), to achieve efficient and convenient data analysis.

Early table understanding methods typically rely on task-specific model architectures trained on specialized datasets Wang et al. (2021); Iida et al. (2021); Nan et al. (2022), which significantly hinders their broader applicability. With the advent of LLMs exhibiting strong generalization capabilities, there is a growing interest in leveraging instruction-tuning techniques to develop table-oriented LLMs capable of handling diverse table-related tasks Zhang et al. (2024a); Li et al. (2024b); Su et al. (2024); Zhang et al. (2024b). These methods have made notable progress in overcoming previous limitations, achieving generalist table models. However, Zheng et al. Zheng et al. (2024) argue that in many real-world scenarios, obtaining high-quality text-based tables is often impractical, whereas table images are more readily available (e.g., screenshots or scanned documents). Moreover, since tables are inherently two-dimensional, image-based representations better preserve their spatial layout and align more closely with human visual intuition. To this end, Zheng et al. Zheng et al. (2024) propose Table-LLaVA, a table MLLM that directly accepts table inputs in image format.

Table-LLaVA demonstrates strong performance across various pre-defined tasks by constructing a large-scale table image dataset, aligning table images and text through pre-training, and applying

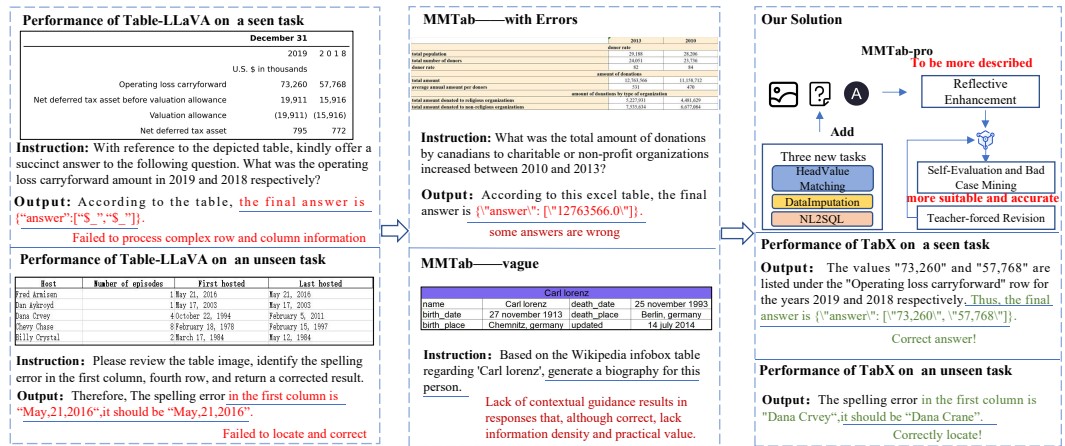

Figure 1: Illustration of the motivation and core design of TabX. Left: Failure cases of Table-LLaVA on complex table layouts and unseen tasks. Middle: Low-quality training samples (e.g., incorrect answers and vague instructions). Right: TabX demonstrates strong performance by constructing a high-quality dataset and a self-evolution with teacher-tuning framework.

supervised instruction tuning. However, through extensive evaluation, we observe that it exhibits limited robustness to complex table layouts and insufficient generalization capabilities for novel tasks. As shown in Figure 1, a common failure case involves tables with hierarchical organization and irregularly merged cells. Furthermore, its performance on unseen tasks often falls short. Through an analysis of the development pipeline of existing table (M)LLMs, we identify two issues: (1) Low-quality instruction-table-answer triplets. As shown in Figure 1, some instructions in existing datasets are vague, and the corresponding answers are overly simplistic. Such ambiguous data fail to guide the model training effectively. (2) Insufficient understanding of table images. Robust performance on both in-distribution and out-of-distribution tasks depends heavily on the model's ability to understand table structure and content. Table-LLaVA falls short in this regard for two main reasons: inappropriate instruction-tuning task settings and a suboptimal choice of foundation model. Effective instruction tuning should involve tasks spanning various levels of difficulty and granularity to promote a deep understanding of the relationships between row and column information. However, Table-LLaVA includes relatively few tasks that require reasoning over global table contents and structures or fine-grained cell-level inference. Moreover, its foundation model, LLaVA Liu et al. (2023a), is exclusively trained on visual understanding tasks, whereas existing research indicates that models trained on both generation and understanding tasks can yield more robust representations Chen et al. (2025a).

To advance table understanding models and promote their broader practical applications, we present **TabX**, a robust and generalizable table MLLM, by addressing the aforementioned challenges. Drawing inspiration from the human reflection and error correction mechanism, the development pipeline of **TabX** comprises two main parts: reflective enhancement-based instruction-tuning dataset construction and self-evolution with teacher-tuning (SETT). In dataset construction, we introduce three additional challenging table understanding tasks to make up for the deficiencies of existing multimodal table datasets in modeling global table semantics and fine-grained cell-level reasoning. The newly constructed instruction-table-answer triplets for these tasks are merged with the existing dataset, expanding the total from 230K to 250K samples. To enhance data quality, we implement a two-step reflective enhancement mechanism, which leverages a teacher model and a student model to reflect on and score the instruction-answer pairs. This process produces triplets with more detailed descriptions, resulting in a high-quality and comprehensive multimodal table dataset, termed MMTab-Pro. Following dataset construction, we design a SETT framework that forms a continuous loop of self-evaluation, teacher-enforced revision, and refined tuning. In this framework, the student model identifies "bad cases" based on self-feedback, while the teacher provides expert guidance to revise them, ensuring the co-evolution of the model and its training data. Moreover, **TabX** is built by fine-tuning Janus-Pro Chen et al. (2025b). This choice is predicated on its superior representational

capabilities, resulting from its joint training on both generation and understanding tasks, allowing for a more comprehensive understanding of table images.

We evaluate **TabX** on the public benchmark MMTab-eval, comparing it against several open-source MLLMs, table-oriented LLMs, and TableLLaVA. Experimental results demonstrate that **TabX** consistently outperforms existing methods across a wide range of table understanding tasks, including both held-in and held-out benchmarks. Significantly, **TabX** exhibits outstanding performance on structurally complex and unseen tasks, highlighting its robustness and generalization capabilities. Extensive ablation studies further validate the effectiveness of each component in contributing to **TabX**'s superior performance.

Our contributions are summarized as follows:

- We introduce **TabX**, a robust and generalizable table MLLMs, achieving new state-of-the-art results across multiple tasks on the MMTab-eval benchmark.

- We introduce three challenging table understanding tasks to complement existing datasets and construct a high-quality instruction-tuning dataset by a reflective enhancement strategy.

- We propose a self-evolution with teacher-tuning framework, enabling collaborative evolution between the model and the data during instruction tuning.

## 2 RELATED WORK

**(M)LLMs for Tabular Tasks.** While LLMs have demonstrated remarkable success across many natural language processing benchmarks, recent studies Bhandari et al. (2024); Dong et al. (2024); Sui et al. (2024) indicate that even the most advanced LLMs may still struggle with complex table-related tasks. This limitation arises from a fundamental modality mismatch: LLMs are primarily trained on one-dimensional textual sequences, whereas tables are inherently two-dimensional. To bridge this gap, recent efforts have introduced fine-tuning strategies tailored specifically for tabular tasks. Early works typically focus on single-task models targeting specific table-related tasks Hegselmann et al. (2023); Andrejczuk et al. (2022); Liu et al. (2021); Kotelnikov et al. (2023); Ren et al. (2025). More recently, research attention has shifted towards developing generalist models. For instance, Li et al. Li et al. (2023c) develop Table-GPT using a new "table-tuning" paradigm. Similarly, Zhang et al. Zhang et al. (2023) propose Table-Llama, a LLaMA-based model fine-tuned via LoRA Hu et al. (2022) on multiple table-related tasks, consistently outperforming its base model across various tasks. On the other hand, Table-Specialist Xing et al. (2024) abandons the one-model-fits-all paradigm in favor of training dedicated specialist LLMs for each tabular task. In contrast, Table-LLaVA Zheng et al. (2024) opts to directly process table images, leveraging their accessibility in real-world settings and naturally preserving 2D structural information. This vision-centric approach has been extended by works like Zhou et al. (2025) and Zhao et al. (2024). The former generates a massive Q&A corpus by prompting an LLM with HTML tables, while the latter adopts a multi-stage fine-tuning strategy on established public datasets. Common to these methods is the use of high-resolution image encoders.

**MLLMs.** Early MLLMs, such as LLaVA Liu et al. (2023b), align visual features extracted by visual foundation models with text embeddings and feed the fused representations into LLMs to facilitate cross-modal understanding between visual and textual content. More recently, there has been a significant push towards developing unified multimodal models that can handle both understanding and generation tasks. For instance, Emu3 Wang et al. (2024) achieves deep multimodal fusion by discretizing heterogeneous data (text, images, videos) into token sequences and processing them through a decoder-only Transformer architecture. Similarly, VILA-U Wu et al. (2024) integrates multimodal understanding and generation within a unified token-based autoregressive framework. Previous methods typically rely on a single visual encoder for both tasks, which often leads to sub-optimal performance due to differing granularity requirements between multimodal understanding and generation. Janus-pro Chen et al. (2025b) addresses this limitation through a decoupled visual encoding strategy. We select Janus-pro as our foundation model to ensure a comprehensive understanding of table images.

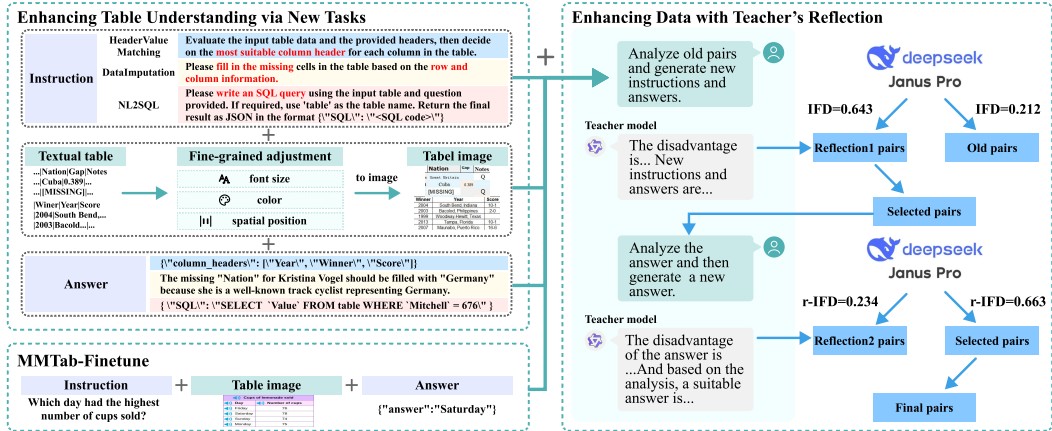

Figure 2: Pipeline of Instruction-tuning Dataset Construction. Three new tasks are integrated and aligned into a unified multi-modal table understanding setting. We then enhance the data quality through a two-stage reflective enhancement, leveraging a teacher model (Qwen-VL-Max) for data revision and a student model for selecting samples that are most beneficial for its learning. Please refer to the appendix for the detailed prompt.

## 3 MMTab-pro: An Expanded Triplet-Structured Multimodal Table Dataset

To facilitate more effective table understanding and address ambiguities in the existing dataset, we construct a new dataset through a two-stage process: (1) integrating additional tasks and (2) reflective enhancement, as shown in Figure 2. We first create instruction-table-answer triplets for three new tasks. Then, we merge them into MMTab-finetune and apply a reflective enhancement strategy to improve the quality of instruction-answer pairs across all triplets.

### 3.1 Enhancing Table Understanding via New Tasks

To enhance the model's comprehensive understanding of tables during training, we introduce additional fine-tuning tasks beyond those originally present in TableLLaVA. These tasks are carefully selected to contain both global structural and semantic understanding, as well as fine-grained cell-level reasoning, promoting a more comprehensive understanding. Specifically, we select three representative tasks: **HeaderValueMatching**, **DataImputation**, and **NL2SQL**. HeaderValueMatching requires the model to determine whether a given cell value in a row correctly belongs to the corresponding column header. DataImputation involves filling in missing values within table cells reasonably. Both of these cell-level reasoning tasks demand the model's ability to locate and understand local context, and correctly interpret row-column semantic relationships. NL2SQL, on the other hand, is a task that transforms natural language questions into structured SQL queries. It requires the model to accurately identify the relevant column names, values, and constraints in the tables, while performing a deep analysis of the tables' structural layout, row–column relationships, and semantic dependencies.

We collect the raw samples for these three tasks from the dataset used by Table-GPT Li et al. (2023c), which consists of instruction-text table-answer triplets. The text-based tables are first converted into Excel format and then augmented using a fine-grained strategy following Table-LLaVA. The resulting tables are rendered as images via screenshots. Moreover, to ensure compatibility with the MMTab-Finetune task format, we standardize the instruction templates and normalize the answer formats. Finally, we merge all samples from the three new tasks with the original MMTab-Finetune dataset, expanding the total number of instruction-table image-answer triplets from 230K to approximately 250K. Detailed descriptions of each task are provided in the appendix.

## 3.2 ENHANCING DATA WITH TEACHER'S REFLECTION

Existing instruction-tuning datasets often suffer from vague instructions and simplistic answers, which limit the model's ability to effectively learn task objectives and generalize across scenarios. To this end, we utilize a two-stage reflection-based data enhancement pipeline Li et al. (2023b; 2024a), focusing respectively on instruction refinement and answer improvement. Specifically, we adopt Qwen-VL-Max Bai et al. (2023) as the teacher model and Janus-Pro Chen et al. (2025b) as the student model. The use of a large teacher model enables more accurate reflection and sample revision, leveraging its stronger reasoning and understanding capabilities. Moreover, we let the student model select the candidate samples to ensure the final data better aligns with the student's requirements. Both models take inputs in the form of <Instruction, Image>.

In the first stage, given an original instruction–answer pair $(x_0, y_0)$, Qwen-VL-Max evaluates the instruction from multiple dimensions, including topical complexity, required specificity, background knowledge, ambiguity, and reasoning difficulty, and generates a new one. Simultaneously, a matching answer is also generated to ensure coherence between the instruction and the answer. Next, we utilize the student model to score both the original pair $(x_0, y_0)$ and the teacher-generated pair $(x_1, y_1)$ using the Instruction-Following Difficulty (IFD) metric Li et al. (2023b). This metric quantifies the contribution of the instruction to the task completion, and a higher IFD indicates that the instruction is more helpful in guiding the model to generate the correct answer. We retain the pair with the higher IFD score as the preferred sample. Taking our chosen pair $(x_1, y_1)$ as an example, in the second stage, the selected pair undergoes answer reflection Li et al. (2024a). Here, the instruction $x_1$ is kept fixed while the teacher model revises only the answer by evaluating its usefulness, relevance, accuracy, and specificity, producing a new instruction-answer pair $(x_1, y_2)$. Subsequently, the student model uses the reversed-IFD (r-IFD) metric to assess whether these two answers contain sufficient information for the model to infer the instruction, with a lower r-IFD being more desirable. The final pair $(x_1, y_1$ or $y_2)$ is thus selected through this two-stage reflective enhancement process. As a result, approximately 80K original samples are replaced with their enhanced versions to form a high-quality instruction-tuning dataset.

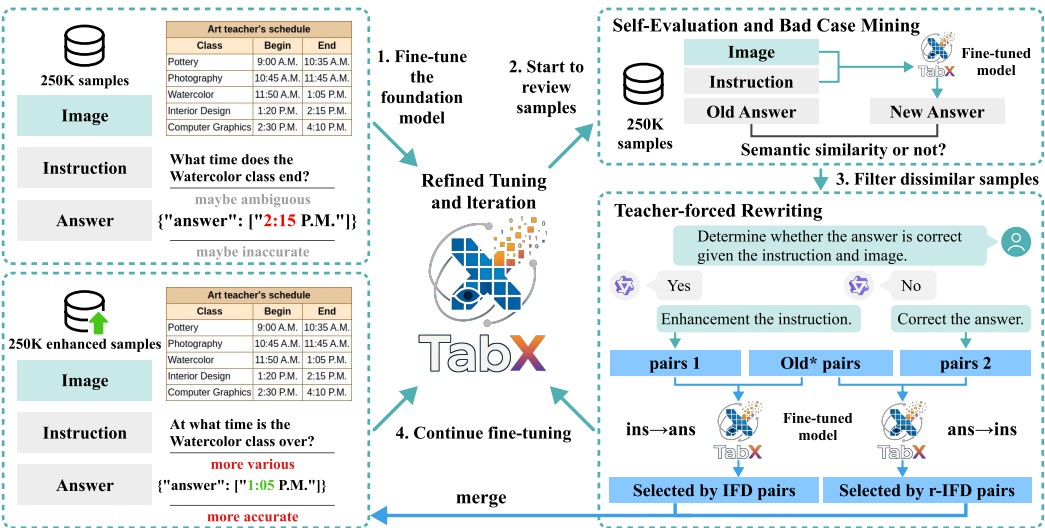

Figure 3: Pipeline of SETT. The process begins with an initial fine-tuning of the student model, followed by its self-evaluation to identify "bad cases". A teacher model then provides expert guidance for revising these samples. The student model is subsequently fine-tuned on the updated dataset, and this cycle is iteratively repeated. Please refer to the appendix for the detailed prompt.

## 4 SETT: SELF-EVOLUTION WITH TEACHER-TUNING

### 4.1 OVERVIEW

While our initial dataset construction employs a powerful teacher model to reflect on and enhance the data, a fundamental challenge persists. A significant capability gap and differing knowledge distributions often exist between large teacher models and smaller student models Li et al. (2025). This discrepancy implies that the teacher's reflective process, and the resulting samples, may not be optimally aligned with the student model's unique learning trajectory. Furthermore, relying on an untrained student model to curate its own data is suboptimal, as it lacks the requisite discernment to accurately identify the most beneficial training instances for its own evolution. To address these challenges, we propose a novel fine-tuning framework, Self-Evolution with Teacher-Tuning, designed to achieve the joint evolution of both the model and its training data by integrating the student's self-feedback and the teacher's expert guidance. As shown in Figure 3, we first fine-tune the student model on the initial dataset to obtain a baseline version. This baseline model then engages in self-evaluation, leveraging its own feedback to identify "bad cases". Subsequently, the teacher model is introduced to guide the revision of these identified samples. Finally, we fine-tune the student model again using the updated dataset, iteratively repeating this process.

### 4.2 SELF-EVALUATION AND BAD CASE MINING

Our SETT introduces a self-evaluation process, which transforms the student model from a passive data consumer into an active agent that pinpoints data-model incongruities. Moreover, this process creates a cycle where improved model capability leads to more precise data curation, which in turn accelerates the model's evolution toward a robust and generalized state. Specifically, for each instruction-table image pair in the training set, we prompt the current fine-tuned student model to generate a corresponding answer $\hat{y}$. We then measure the semantic alignment between the model's prediction $\hat{y}$ and the ground-truth answer $y$. A high degree of misalignment indicates a potential "bad case". To quantify this alignment, we compute the cosine similarity of their semantic embeddings. Drawing from the findings of Zhao et al. (2025), which suggest that average or max pooling over all token embeddings is more effective for capturing overall semantics than relying on special tokens (e.g., the first or last token) in encoder–decoder architectures, we implement the following procedure: (1) We encode both $\hat{y}$ and $y$ using the student model to obtain the hidden states of all tokens. (2) We apply average pooling over the token-level hidden states to derive semantic embeddings. (3) We compute cosine similarity between the semantic embeddings of $\hat{y}$ and $y$. If the similarity score falls below a predefined threshold $\delta$, the sample is flagged as a "bad case". These samples are passed to the next stage of our SETT framework for teacher-forced revising. We experimentally set the threshold $\delta$ to 0.5.

### 4.3 TEACHER-FORCED REVISION

Following the identification of "bad cases", the framework introduces the teacher model to perform a crucial diagnostic and corrective function. The teacher acts as an external and more knowledgeable expert, providing the necessary guidance to resolve these "bad cases" and ensure the student's evolutionary path remains productive. For each "bad case", the teacher model determines the correctness of the original answer $y$ given the table image and instruction. If $y$ is identified as erroneous, i.e., it is a truly misleading sample, the teacher' role is correction. It is prompted to generate a revised, high-quality answer. Conversely, if $y$ is validated as correct, the incongruity is attributed to the student's misinterpretation, likely stemming from a non-robust or ambiguous instruction. Here, the teacher's role shifts to clarification. We prompt the teacher to rephrase the instruction into a more robust and semantically equivalent form. Crucially, the teacher's revision is not unconditionally accepted. The modified sample is subsequently passed back to the student model for a final verification using the IFD and r-IFD metrics. This step upholds the principle of student-centric learning, ensuring that the teacher's guidance is indeed beneficial from the student's current perspective.

### 4.4 REFINED TUNING AND ITERATION

Following the teacher-forced revision, the newly enhanced samples are merged with the set of high-quality samples retained from the self-evaluation phase. This updated dataset is then used to initi-

| Task Types | Table Structure Understanding (TSU) | | | | | | | Academic Tabular Tasks | | | | | | | | | |
|---|---|---|---|---|---|---|---|---|---|---|---|---|---|---|---|---|---|
| | | | | | | | | TQA | | | | TFV | | | T2T | | |
| Benchmarks | TSD | | TCL | RCE | | MCD | TCE | TR | TWP | WTQ | HiT | TAT | TF | IT | FT | HiT_t2t | RW | WI | TO |
| Methods | Row | Col. | Acc. | RowF1 | ColF1 | F1 | Acc. | AT | Acc. | Acc. | Acc. | Acc. | Acc. | Acc. | BLUE | BLUE | BLUE | BLUE | BLUE |
| BLIP2 | 0.2 | 0.0 | 0.1 | 0.0 | 0.0 | 0.0 | 0.1 | 0.1 | 3.4 | 2.1 | 1.5 | 2.2 | 18.7 | 27.5 | 2.3 | 2.6 | 1.1 | 0.7 | 4.3 |
| Qwen-VL | 0.0 | 0.0 | 0.0 | 0.0 | 0.4 | 0.0 | 0.0 | 0.8 | 3.3 | 0.1 | 0.1 | 0.1 | 1.1 | 0.7 | 0.5 | 0.2 | 0.1 | 0.0 | 0.8 |
| VILA-U | 0.1 | 4.0 | 0.3 | 4.1 | 10.5 | 3.0 | 1.1 | 27.0 | 17.9 | 4.5 | 1.2 | 4.4 | 10.9 | 14.2 | 6.8 | 0.7 | 1.6 | 3.5 | 7.5 |
| LaVIT | 0.1 | 3.3 | 0.1 | 2.0 | 9.4 | 2.0 | 0.4 | 29.5 | 13.4 | 3.2 | 0.9 | 3.8 | 6.3 | 12.9 | 5.2 | 0.7 | 1.2 | 3.4 | 7.0 |
| Janus-pro | 1.3 | 4.2 | 0.3 | 4.8 | 11.7 | 3.1 | 1.7 | 31.3 | 18.3 | 4.2 | 1.3 | 5.3 | 10.5 | 22.1 | 9.7 | 1.0 | 1.4 | 2.1 | 9.6 |
| Table-LLaMA +OCR | 3.9 | 3.6 | 6.5 | 2.8 | 2.4 | - | 4.0 | - | 11.1 | 12.5 | 13.5 | 2.7 | 44.5 | 2.2 | 25.4 | 0.1 | 0.1 | 0.3 | - |
| Table-LLaVA | 33.1 | 33.2 | 29.3 | 31.0 | 37.9 | 17.1 | 19.4 | 47.0 | 57.8 | 18.4 | **10.1** | 12.8 | **59.8** | **65.0** | 25.6 | **9.7** | **10.5** | 9.7 | 23.0 |
| Ours | **37.4** | **60.8** | **35.6** | **33.9** | **47.0** | **31.9** | **26.8** | **54.2** | **81.1** | **18.6** | 9.3 | **32.4** | 52.5 | 59.7 | **26.1** | 7.1 | 4.6 | **9.8** | **23.6** |

Table 1: Comparison of the performance on the held-in benchmarks. "TWP", "HiT", "TAT", "TF", "IT", "FT", "HiT_t2t", "RW", "WB", and "TO" correspond to "TABMWP", "HiTab", "TAT-QA", "TabFact", "InfoTabs", "FeTaQA", "HiTab_t2t", "Rotowire", "WikiBIO", and "ToTTo", respectively. "AT" denotes the average TEDS score for recognizing table formats in HTML, Markdown, and LaTeX. Detailed definitions of each task are provided in the appendix. The best results are highlighted in bold.

ate the next round of fine-tuning for the student model. This iterative loop of teacher-forced self-evolution empowers the model to progressively refine its own data curriculum as its capabilities advance. Ultimately, this leads to the development of **TabX**, a robust and generalized table MLLM.

## 5 EXPERIMENTS

### 5.1 EXPERIMENTAL SETTINGS AND RESULTS

**Datasets and Evaluation Metrics.** Following Zheng et al. (2024), we conduct all experiments on the publicly available multi-modal table image benchmark dataset MMTab-eval. This dataset contains a wide range of table-based question answering tasks and visual reasoning challenges, offering strong task diversity and difficulty. We adopt different evaluation metrics based on task types. For the Table Question Answering (TQA), Table Fact Verification (TFV), and Table-to-Text (T2T) tasks, we use Accuracy or BLEU score Papineni et al. (2002) as the evaluation metrics. For the Table Size Detection (TSD) task, we calculate the accuracy of the predicted number of rows and columns. For the Table Cell Extraction (TCE) and Table Cell Locating (TCL) tasks, we use cell-level accuracy as the measurement standard. For the table recognition (TR) task, we use the Tree-Edit-Distance-based Similarity (TEDS) score Zhong et al. (2020). For the Merged Cell Detection (MCD) task, we evaluate model performance using cell-level F1 score. For the Row&Column Extraction (RCE) task, we compute the cell-level F1 score for both row-wise and column-wise extraction results Zheng et al. (2024).

**Baselines.** We compare our method against (1) open-source general-purpose MLLMs such as Janus-Pro Chen et al. (2025b), Qwen-VL Bai et al. (2023), BLIP-2 Li et al. (2023a), VILA-U Wu et al. (2024), and LaVIT Jin et al. (2023); (2) table-oriented LLMs such as TableLLaMA Zhang et al. (2023); and (3) Table-LLaVA. All compared models are 7B in size for a fair comparison.

**Implementation Details.** In our pipeline, the teacher model is Qwen-VL-Max (prompts are detailed in the appendix), which is selected for its powerful vision-language capabilities and cost-effectiveness, with the entire process costing approximately $200. The student model is the 7B version of Janus-Pro. All experiments are conducted on four NVIDIA A800 GPUs. We adopt the LoRA efficient fine-tuning strategy, use the AdamW optimizer with an initial learning rate of 2e-5, and apply linear warm-up followed by cosine decay scheduling. During fine-tuning, self-evaluation and teacher-forced revision are conducted every two training epochs. This entire process is iterated for a maximum of three times, terminating early if no "bad cases" are identified during the self-evaluation stage.

**Results on the Held-in Benchmarks.** The held-in benchmarks include 17 tasks, primarily categorized into academic tabular tasks and table structure understanding tasks. Please refer to the appendix for dialogue process visualizations. As shown in Table 1, general-purpose multimodal

| Method | TSD | | RCE | | TCL | TCE | AIT | PHT | TCQ |
|---|---|---|---|---|---|---|---|---|---|
| | Row | Col. | RF1 | CF1 | Acc. | Acc. | Acc. | Acc. | Acc. |
| VILA-U | 1.4 | 4.1 | 0.1 | 2.6 | 0.1 | 2.0 | 2.3 | 13.7 | 10.0 |
| LaVIT | 0.8 | 2.3 | 0.8 | 1.9 | 0.1 | 1.9 | 1.7 | 8.5 | 9.8 |
| Janus-pro | 1.6 | 4.4 | 12.1 | 3.2 | 0.3 | 2.2 | 4.6 | 15.1 | 10.9 |
| Table-LLaVA | 25.2 | 16.4 | 22.0 | 18.1 | 26.1 | 11.3 | 5.4 | 51.0 | **44.0** |
| Ours | **31.6** | **47.7** | **28.7** | **34.7** | **31.1** | **24.6** | **6.9** | **52.9** | 40.5 |

Table 2: Comparison of the performance on the held-out benchmarks. "RF1" denotes the F1 score for row prediction, and "CF1" denotes the F1 score for column prediction. "AIT", "PHT", and "TCQ" correspond to "AIT-QA", "PubHealthTab", and "TabMCQ".

models perform poorly, mainly due to the lack of specific training on table images. Among them, Janus-Pro performs best, benefiting from joint training on generation and understanding tasks, which helps capture more robust visual representations. This motivates our choice of Janus-Pro as the foundation model for **TabX**. Table-LLaMA outperforms general MLLMs due to specialized tabular training. However, it suffers from OCR inaccuracies and the loss of structural information. Table-LLaVA, designed specifically for table image understanding tasks, significantly outperforms previous methods on many tasks. Nonetheless, it shows mediocre performance on complex table structure and semantic reasoning tasks (e.g., MCD and HiTab QA). In contrast, **TabX** consistently excels across almost all tasks, with particularly notable gains on complex tasks such as TCE, MCD, tabular numerical reasoning (TABMWP and TAT-QA), and HiTab QA. This superior performance stems from our introduction of three complex tasks for fine-tuning, which guide the model toward a deeper comprehension of table content and structure, as well as the continuous enhancement of data quality through reflection mechanisms and the proposed SETT framework. Notably, **TabX** underperforms on the Rotowire (RW) dataset. Our qualitative analysis (see the appendix) indicates this is not a failure of comprehension but rather a limitation of the evaluation metric. We find that **TabX** excels at extracting a higher density of factual information tables. Conversely, Table-LLaVA's outputs, though less factually grounded, align better stylistically with the reference answers, thereby achieving a higher BLUE score. This suggests a potential misalignment between n-gram-based scores and true factual accuracy on this task.

**Results on the Held-out Benchmarks.** Following Table-LLaVA, we validate our method on held-out data not present in the training set. As shown in Table 2, we observe similar trends. It is worth noting that general-purpose MLLMs are less affected by unseen data, as they are not trained specifically on table images and primarily rely on their own generalization capabilities. Compared to Table-LLaVA, our model maintains its advantage across multiple held-out settings. This is attributed to our strategy of setting appropriate tasks and enhancing data quality, which unleashes the model's generalization capabilities.

| Method | TQA | TFV | TSU | T2T | Held-out |
|---|---|---|---|---|---|
| w/o T | 30.5 | 39.7 | 36.0 | 10.7 | 25.4 |
| w/o R | 28.5 | 38.7 | 36.5 | 10.4 | 22.1 |
| w/o S | 24.7 | 38.2 | 32.9 | 10.2 | 22.0 |
| Ours | **35.4** | **46.0** | **40.9** | **11.3** | **33.2** |

Table 3: Ablation studies. We report the average performance on the TQA, TFV, T2T, TSU, and Held-out tasks. "T": three new tasks; "R": reflection-based data enhancement; "S": SETT.

## 5.2 Ablation Study

To evaluate the effectiveness of each key component in **TabX**, we conduct ablation experiments by individually removing: (1) the three new tasks, (2) the reflection-based data enhancement pipeline, and (3) the SETT framework. Results are reported in Table 3. As shown in the table, removing the three new tasks leads to notable performance drops. This highlights the importance of the three tasks in promoting the model's ability to handle structurally and semantically challenging tables. Furthermore, comparing the performance with and without the reflective enhancement mechanism reveals approximately 5% to 10% improvement on many tasks, indicating the effectiveness of enhancing triplet quality. Notably, the removal of the SETT framework results in the most substantial

performance degradation across all benchmarks. This result provides compelling evidence for the effectiveness of SETT, which enables a co-evolution between the model's capabilities and the data quality.

## 6 DISCUSSION

**Robustness to Structural Perturbations.** To ensure **TabX** comprehensively understands table structure and content, we specifically introduce three challenging fine-tuning tasks that demand both global structural and semantic understanding, as well as fine-grained cell-level reasoning. In addition, we apply data quality enhancement techniques to ensure effective training. To assess whether **TabX** truly understands table structure and content, we design a perturbation test. Specifically, we select 400 table images from the test set and randomly permute the structure of each by swapping either two rows or two columns (the first row/column is excluded), while keeping the instruction unchanged. As shown in Table 4, **TabX** remains robust under this structural perturbation. For comparison, we also evaluate Janus-Pro and Table-LLaVA on the same 400 samples. Notably, Janus-Pro produces inconsistent results in nearly every swapped case, while Table-LLaVA exhibits inconsistencies on more samples than our method. A model that fully understands table structure and semantics should be robust to column-swapping operations that do not affect the overall table semantics. In contrast, models that only perceive local table content or are insensitive to row-column relationships would fail in such scenarios.

| Method | TQA | TFV | MEC-EL | MEC-EC | CTC |
|---|---|---|---|---|---|
| Janus-pro | 0.15 | 0.07 | - | - | - |
| Table-LLaVA | 0.90 | 0.88 | 0.03 | 0.01 | 0.05 |
| Ours | **0.94** | **0.92** | **0.15** | **0.14** | **0.31** |

Table 4: Performance on perturbation test and new tasks. "EL" and "EC" stand for Error Location and Error Correction, respectively. The best results are highlighted in bold.

**Generalization to Unseen Tasks.** We further explore **TabX**'s ability to generalize to unseen tasks, rather than merely unseen data within known tasks (held-out benchmarks). Specifically, we design two new tasks: (1) Misspelled Entry Correction (MEC), where the model must locate a spelling error and provide the correct version; and (2) Cell Type Classification (CTC), which involves identifying a cell's type (e.g., header, data, merged) from its coordinates. We construct a dataset of 400 samples for each of these tasks. The evaluation metric is accuracy, reflecting the fraction of tasks completed successfully. As shown in Table 4, for the MEC task, **TabX** achieves scores of 0.15 in error location and 0.14 in error correction, substantially higher than Table-LLaVA's 0.03 and 0.01. Similarly, in the CTC task, **TabX**'s accuracy of 0.31 is significantly better than the 0.05 achieved by Table-LLaVA. These results suggest that when instruction-tuning tasks are carefully designed and paired with high-quality data, the model can learn meaningful alignments between table images, instructions, and answers. This, in turn, allows it to better leverage the generalization capacity of MLLMs and adapt to new tasks more effectively.

## 7 CONCLUSION

In this paper, we introduce **TabX**, a robust table MLLM designed to address the limited robustness and generalization capabilities of existing table understanding models. We identify two critical issues in the current table (M)LLM development pipeline: low-quality instruction-table-answer triplets and insufficient table image comprehension. To overcome these issues, we first improve the existing instruction-tuning dataset by introducing three challenging fine-tuning tasks to foster a deeper understanding of table context and using a reflective enhancement to boost data quality. We then propose a self-evolution with teacher-tuning framework, which adaptively optimizes training data based on the model's own feedback. Extensive experiments on the MMTab-eval benchmark demonstrate that **TabX** consistently outperforms existing methods. Notably, **TabX** exhibits exceptional robustness and generalization, particularly on structurally complex and unseen tasks.

ETHICS STATEMENT

We have considered the ethical implications of this work. Our model is trained on established public datasets and a new dataset we synthesized by rendering tables from a public text corpus. Besides, while the data's English focus may create biases, our work is intended solely for the positive application of advancing automated document understanding.

REPRODUCIBILITY STATEMENT

We are committed to making our work reproducible. Core components of our source code and a sample of our synthesized dataset are provided in the supplementary materials. This partial code is also anonymously hosted at: https://anonymous.4open.science/r/tabx-F058/. Further details on hyperparameters and our experimental methodology are described in Section 5 and the appendix. We commit to releasing the full codebase, dataset, and pre-trained models publicly upon the paper's acceptance to facilitate future research.

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

## A    THE USE OF LARGE LANGUAGE MODELS (LLMS)

We used a large language model (LLM) solely as a tool for grammar correction and language polishing during the preparation of this manuscript. All authors have reviewed the final text and assume full responsibility for its content and scientific integrity. This use as a writing assistant is distinct from the LLM integrated into our research methodology, which is the subject of this paper.

# B  IMPLEMENTATION DETAILS OF TABX

## B.1  THE PROMPTS USED IN DATASET CONSTRUCTION

To enhance triplet quality during dataset construction, we employ a two-stage reflection-based data enhancement pipeline. Figure 4 presents the prompts used by the teacher model in both stages.

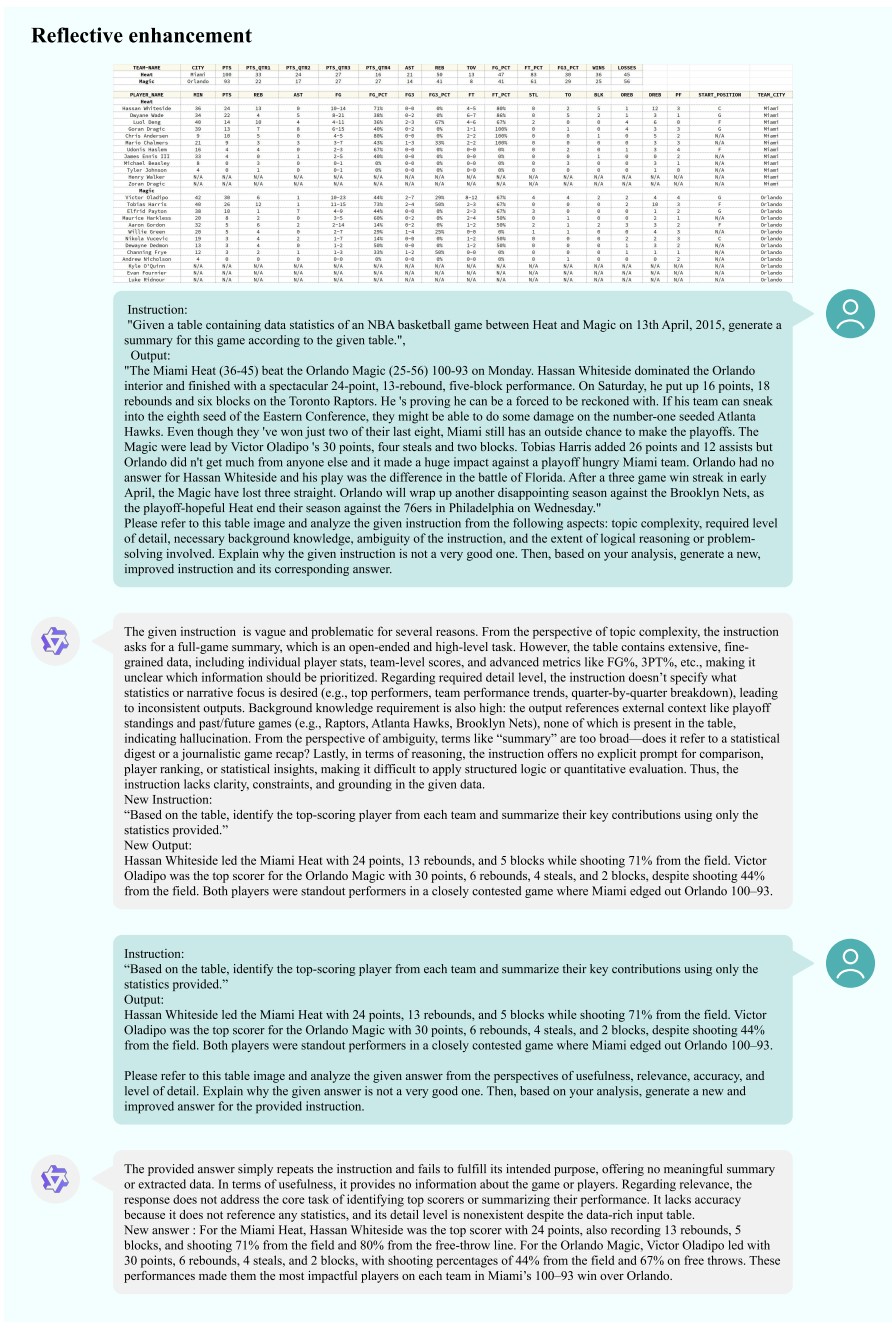

Figure 4: Prompts used in dataset construction.

## B.2 DEFINITION OF IFD AND R-IFD

In the aforementioned pipeline, the student model utilizes two metrics, Instruction-Following Difficulty (IFD) and reversed-IFD (r-IFD), to select samples most beneficial for its own learning. The definitions of IFD and r-IFD are as follows:

$$\text{IFD}_\theta(y|x) = \frac{\text{ppl}(y|x)}{\text{ppl}(y)} = \exp\left(L_\theta(y|x) - L_\theta(y)\right), \tag{1}$$

$$\text{r-IFD}_\theta(x|y) = \frac{\text{ppl}(x|y')}{\text{ppl}(x)} = \exp\left(L_\theta(x|y') - L_\theta(x)\right). \tag{2}$$

Here, $L_\theta(\cdot|\cdot)$ denotes the cross-entropy loss computed by the model under different contextual conditions during inference, which can be used to derive perplexity scores. Building on this, IFD quantifies the helpfulness of an instruction $x$ in guiding the model to generate a target response $y$ by comparing the model's perplexity in predicting $y$ with and without the instruction. A lower perplexity conditioned on the instruction indicates that the instruction provides meaningful guidance. In contrast, r-IFD assesses the informativeness of a response $y$ in implying its original instruction $x$. To achieve this, $y$ is rephrased into a query-like form $y'$, designed to "guess" the missing instruction, and the model's perplexity in reconstructing $x$ from $y'$ is compared against its unconditional generation of $x$.

## B.3 THE PROMPTS USED IN TEACHER-FORCED REVISION

In our proposed SETT framework, the teacher model is employed to determine the correctness of the original answer given the instruction and table image. If the teacher model identifies an error, it corrects the answer. Otherwise, the teacher model is prompted to rephrase the instruction. Figure 5 and 6 show the prompts used for error correction and clarification, respectively.

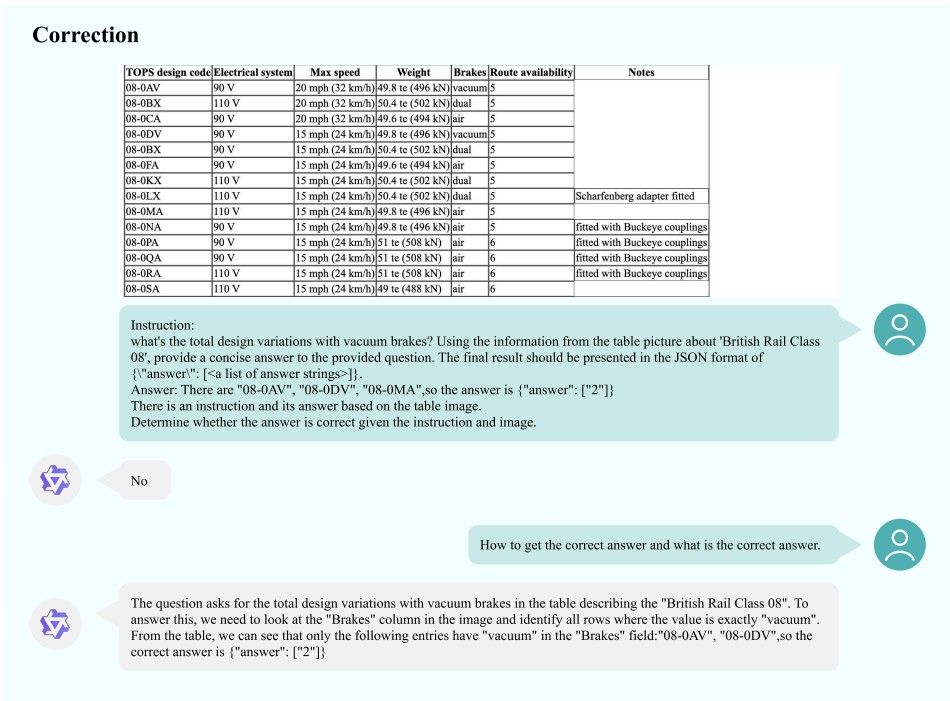

Figure 5: Prompts used for correction.

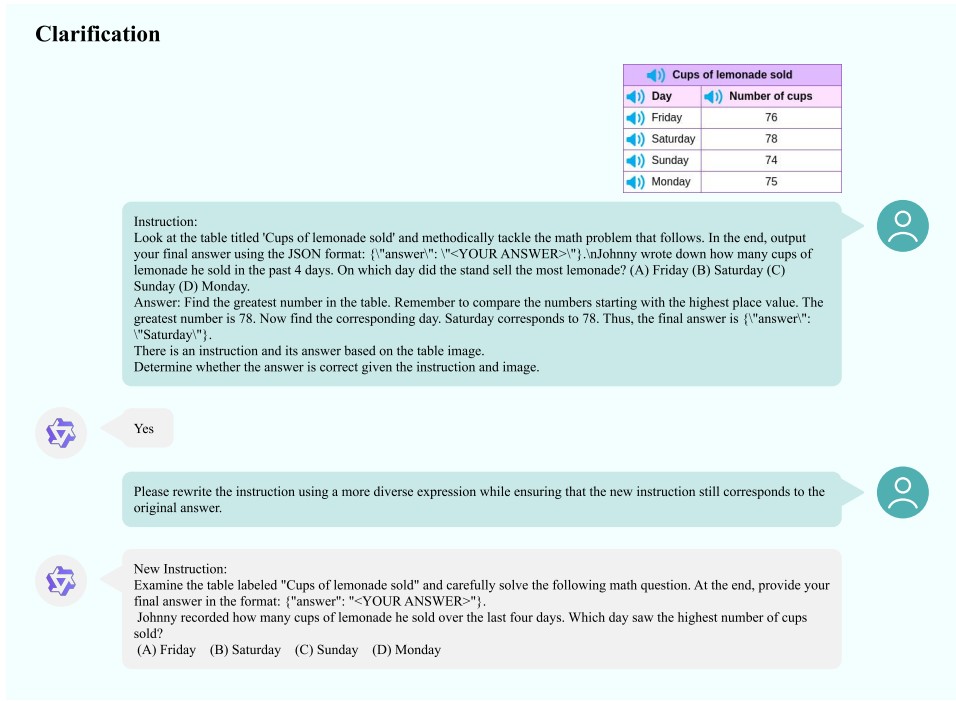

Figure 6: Prompts used for clarification.

## C EXTENDED ANALYSIS AND VISUALIZATIONS

### C.1 TASK DEFINITIONS

We show the definition of each task in Table 5.

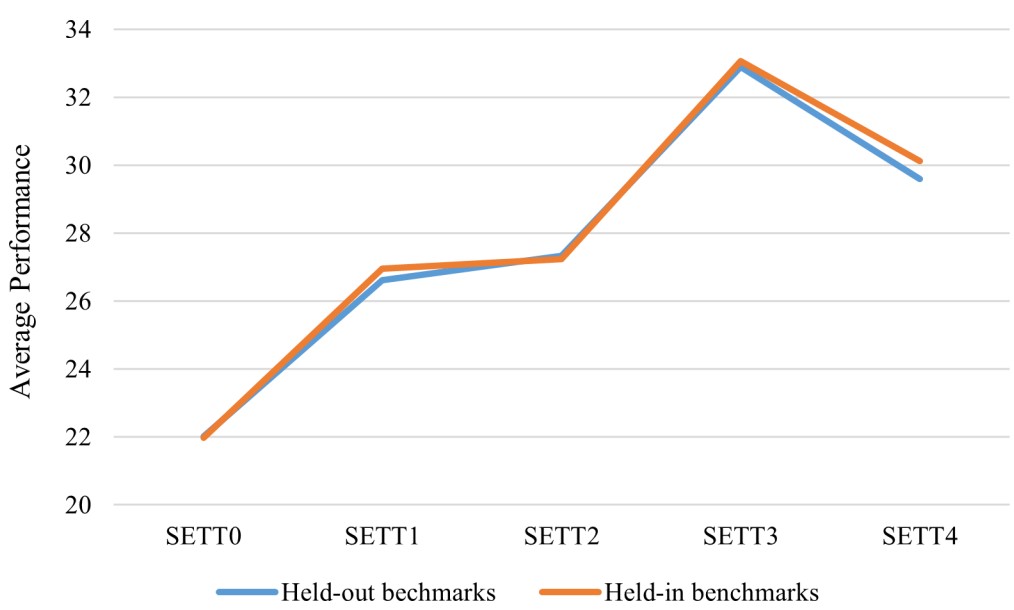

Figure 7: Model performance across iterative SETT rounds.

## C.2 IMPACT OF THE NUMBER OF ITERATIONS

A critical consideration within our framework is the optimal number of fine-tuning iterations. To explore this, we conduct an experiment tracking model performance over four consecutive rounds. As shown in Figure 7, the first round leads to a significant performance improvement, followed by diminishing returns in the second and third iterations. Notably, after the third cycle, we observe a performance degradation. This indicates a delicate balance exists, where excessive iteration can transform beneficial guidance into a source of noise and stylistic bias.

## C.3 DIALOGUE VISUALIZATIONS

### C.3.1 VISUALIZATION RESULTS ON SEEN TASKS.

Figure 8 and 9 showcase a series of dialogue visualizations on both simple and complex tables. For simple tables, as shown in Figure 8, our method consistently demonstrates superior performance in precise information localization and extraction. In Case 1, our method reliably extracts structural metadata and adheres to specific answer formats, while Janus-Pro and Table-LLaVA exhibit inconsistent instruction following and less accurate recognition. In Case 2 and Case 4, our model accurately identifies the table structure to pinpoint the exact cell at the intersection of specified rows and columns. This contrasts with Table-LLaMA's limitations due to OCR-induced loss of relational context, Janus-Pro's frequent misidentification of relevant rows, and Table-LLaVA's tendency to generate irrelevant or non-existent content. Case 3 highlights our model's ability to perform multi-step information retrieval and computation by accurately extracting and comparing values from different years. Conversely, Table-LLaMA leads to year-value misalignment, and both Janus-Pro and Table-LLaVA often extract only a single value without performing the required comparison.

As shown in Figure 9, the performance gap widens on complex tables (Cases 5-6), which demand a deeper understanding of hierarchical relationships and comparative reasoning. Here, the competitors struggle to navigate multi-level headers to find the correct entry (Case 5) or fail to comprehend instructions requiring comparisons across columns, such as identifying the "highest relative increase" (Case 6), often resorting to hallucination. **TabX**, however, successfully resolves these intricate queries.

### C.3.2 RESULTS ON COLUMN-SWAP TEST AND UNSEEN TASKS

**Column-swap Test.** We evaluate the model's robustness against structural perturbations by swapping two columns in the tables. As shown in Figure 10, this simple modification induces significant failures for Table-LLaVA, whose outputs become inconsistent and remain incorrect across the swaps. In contrast, **TabX**'s performance is unaffected by this operation.

**Unseen Tasks.** Figure 11 shows the results of two unseen tasks. In the MEC task, Table-LLaVA fails to locate and correct the error. In the CTC task, it completely disregards the classification directive, merely extracting the cell's value instead. Conversely, **TabX** successfully executes both novel instructions.

## C.4 VISUALIZATION OF THE ENHANCEMENT PROCESS

In this section, we present the evolutionary trend of the instruction-tuning triplets and the corresponding changes in inference results under the proposed SETT framework. From the visualizations in Figure 12, we observe the following phenomena. As training progresses, the model's responses progressively transform from being vague or irrelevant to accurate, complete, and structurally coherent. This progression demonstrates that through the iterative cycle of self-evaluation, teacher-forced revision, and fine-tuning, the student model effectively learns to correct its initial response biases, significantly improving its understanding of table images and its ability to accomplish associated tasks. Simultaneously, we observe an increasing richness and precision in instruction content. While instructions in the initial stages are largely templated or simplified, the teacher model can generate information-dense instructions in subsequent iterations. These instructions include multi-step reasoning, integration of ambiguous information, and comparative analysis across table rows and columns.

## C.5 Qualitative Analysis on the Rotowire Dataset

We present two examples from the Rotowire Dataset in Figure 13 and 14 to analyze the performance gap. To facilitate comparison, we mark corrected identified facts in red and incorrect facts in blue. In both cases, **TabX** demonstrates superior factual grounding compared to Table-LLaVA. For the first example, **TabX** correctly references 9 out of 12 mentioned facts, while Table-LLaVA only manages 7 out of 11. This trend is amplified in the second example, where **TabX** correctly identifies 20 of 41 factual elements, significantly outperforming Table-LLaVA's 12 correct out of 32. However, in both cases, Table-LLaVA achieves a higher BLUE score.

| Name | Task Category | Task Name | Dataset | Task Description |
|------|---------------|-----------|---------|-----------------|
| MMTab-pro | Question Answering | Flat TQA (F TQA) | WTQ | TQA based on tables with flat structure and a single-row header. |
| | | Free-form TQA | FeTaQA | TQA with a free-form text answer rather than a copied span from the table. |
| | | Hierarchical TQA (H TQA) | HiTab | TQA based on tables with complex structures, including multi-level (hierarchical) headers and merged cells that span multiple rows or columns. |
| | | | AIT-QA | |
| | | Multi-choice TQA | TabMCQ | TQA with multiple-choice questions. |
| | | Tabular Numerical Reasoning | TABMWP | TQA requiring math reasoning like max/min or computations. |
| | | | TAT-QA | |
| | | HeaderValueMatching | Synthesis | TQA requiring aligning table headers with corresponding cell values to enhance structural understanding. |
| | | DataImputation | Synthesis | TQA requiring filling in missing table cells based on observed values and table semantics. |
| | Fact Verification | Table Fact Verification | TabFact | Determine the factual consistency between a table and a given statement by predicting whether the statement is supported or refuted by the tabular evidence. |
| | | | InfoTabs | |
| | | | PubHealthTab | |
| | Text Generation | Cell Description | ToTTo | Generate a one-sentence description for the highlighted table cells, with some tasks additionally providing explicit operations (e.g., SUM, AVERAGE) to guide the generation. |
| | | | HiTab_T2T | |
| | | Game Summary | Rotowire | Generate a detailed NBA game summary based on tables containing box and line scores, with reference summaries sourced from Rotowire. |
| | | Biography Generation | WikiBIO | Generate biography from personal information table. |
| | | NL2SQL | Synthesis | Generate SQL query from natural language question and table schema. |
| | Structure Understanding | Table Size Detection | TSD | Determine table's row and column count. |
| | | Table Cell Extraction | TCE | Extract text from specified (row, column) locations. |
| | | Table Cell Locating | TCL | Find (row, col) position of given cell values. |
| | | Merged Cell Detection | MCD | Detect merged cells and return bounding positions. |
| | | Row&Column Extraction | RCE | Extract all cells from specified rows/columns. |
| | | Table Recognition | TR | Convert table image to HTML/Markdown/Latex format. |

Table 5: Detailed descriptions of the evaluation tasks, including their abbreviations, full names, and corresponding task definitions.

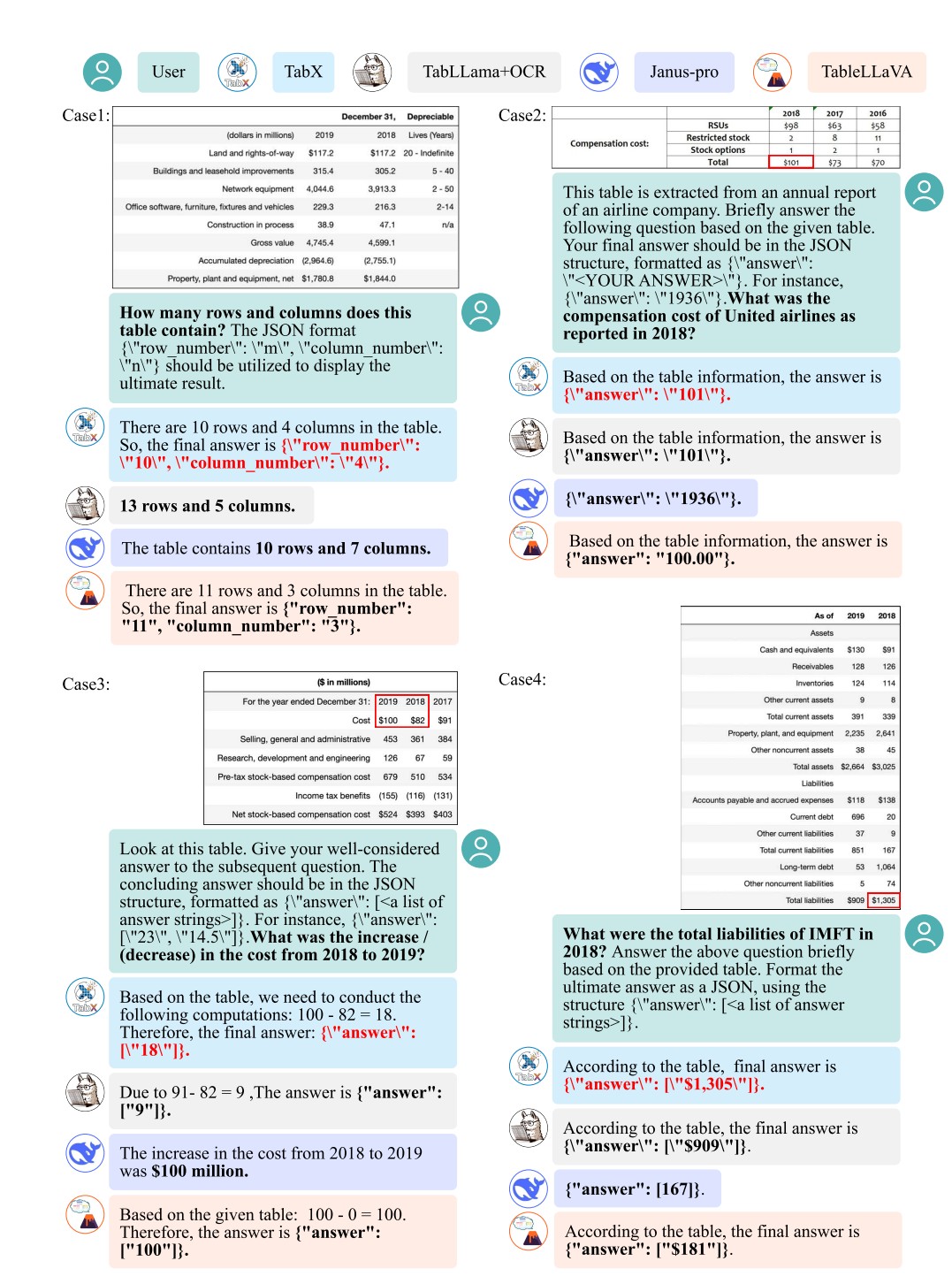

Figure 8: Dialogue visualizations on simple tables.

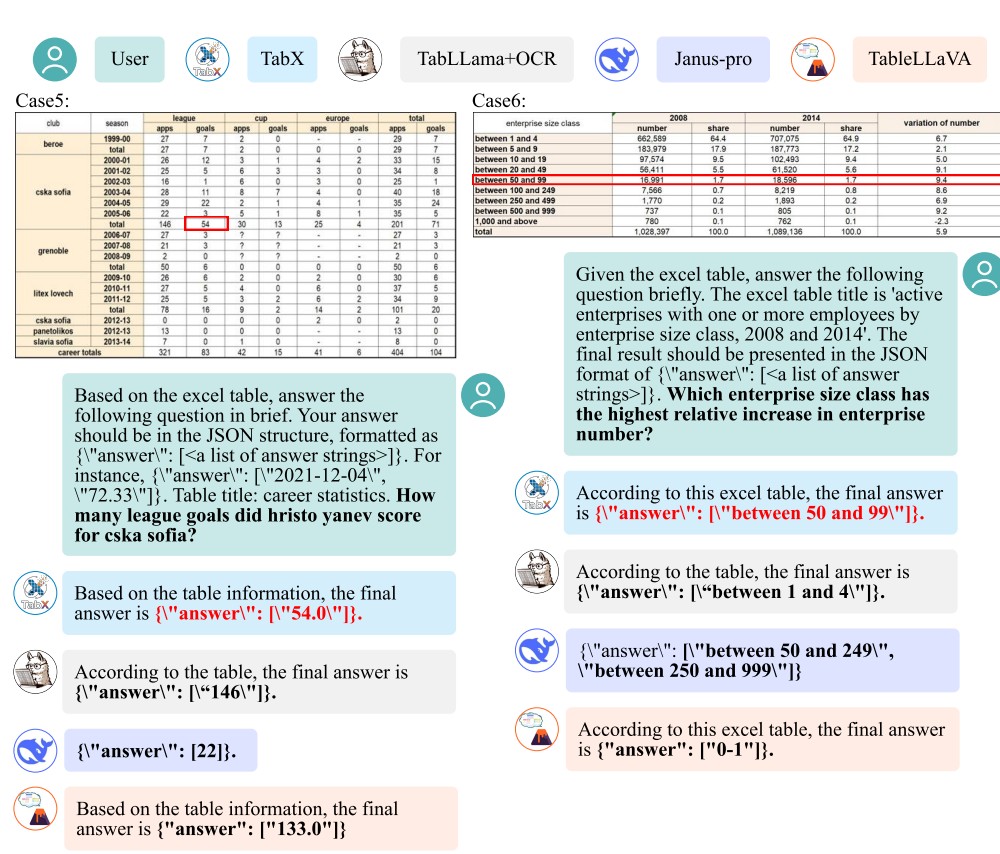

Figure 9: Dialogue visualizations on complex tables.

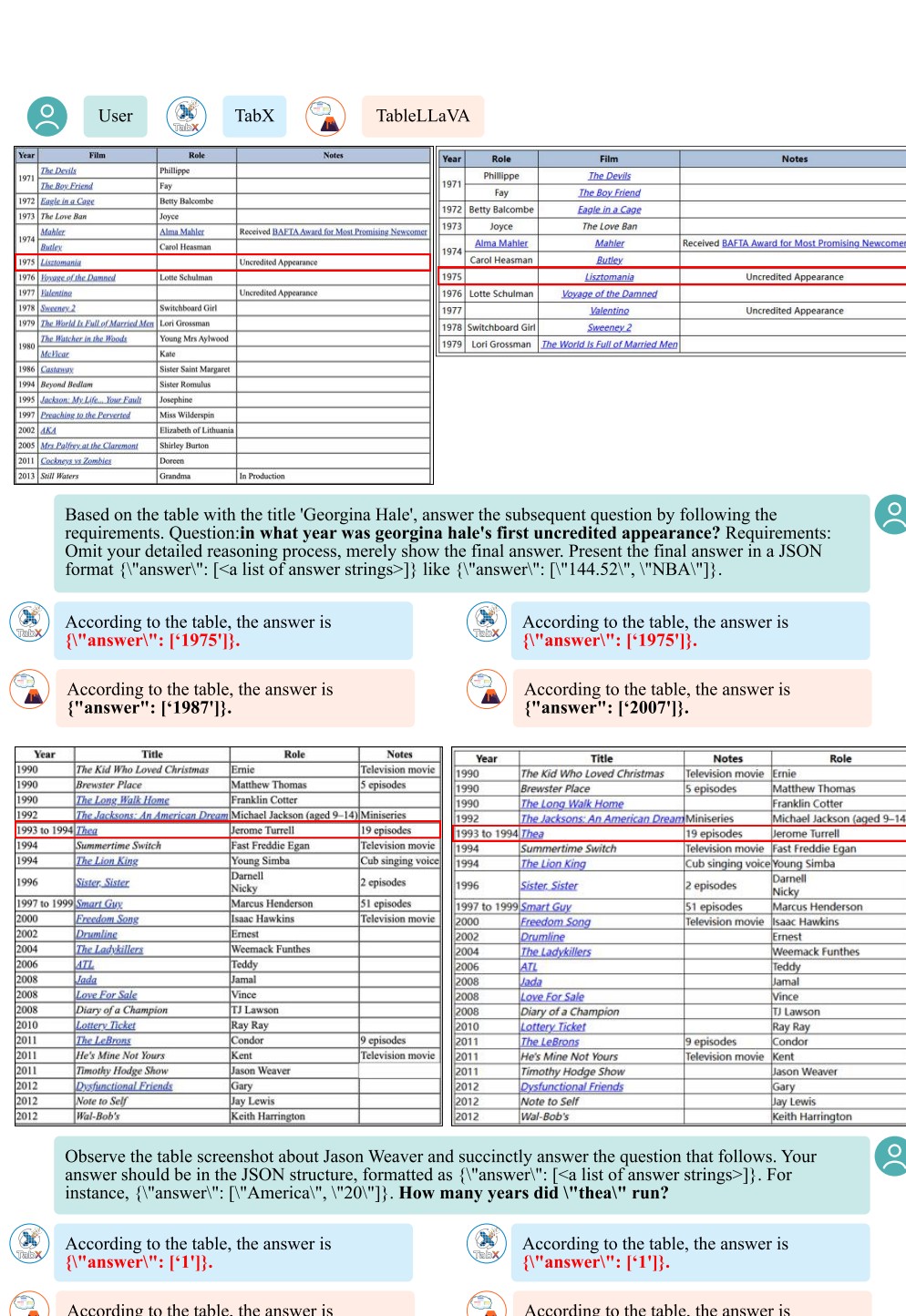

Figure 10: Visualization of model predictions under the column-swap test.

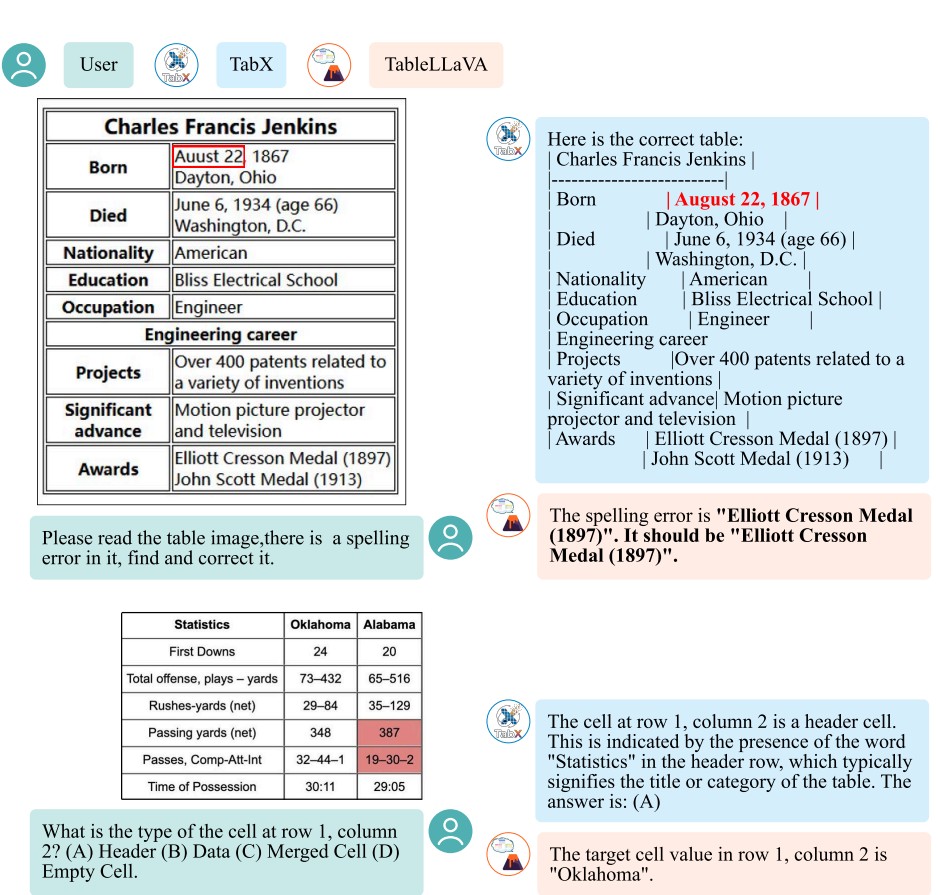

Figure 11: Dialogue visualization on unseen tasks.

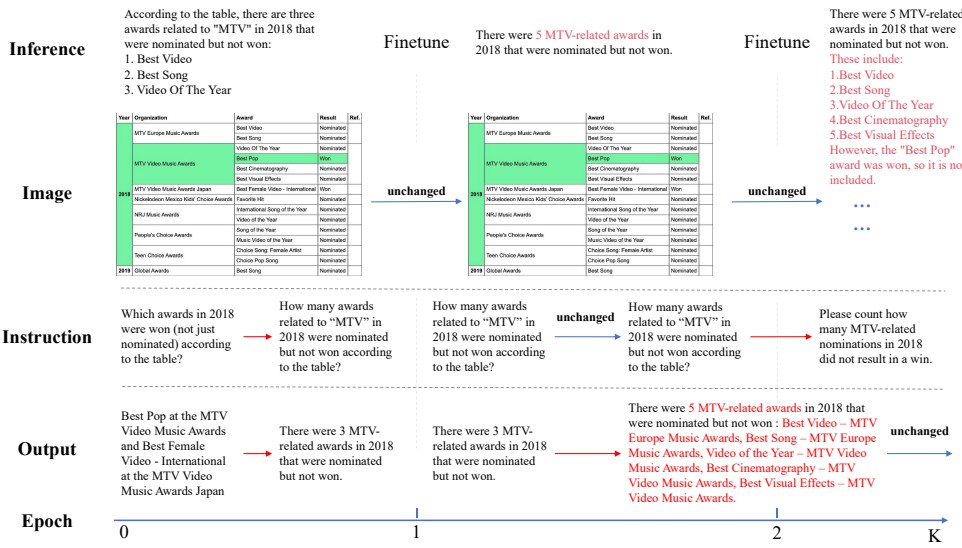

Figure 12: Visualization of the enhancement process. "Inference" denotes the prediction of **TabX**.

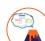

**User**    **TabX**    **TableLLaVA**

| TEAM-NAME | CITY | PTS | PTS_QTR1 | PTS_QTR2 | PTS_QTR3 | PTS_QTR4 | AST | REB | TOV | FG_PCT | FT_PCT | FG3_PCT | WINS | LOSSES | | | |
|---|---|---|---|---|---|---|---|---|---|---|---|---|---|---|---|---|---|
| Jazz | Utah | 95 | 24 | 23 | 27 | 21 | 16 | 44 | 11 | 44 | 86 | 29 | 6 | 17 | | | |
| Heat | Miami | 100 | 29 | 33 | 16 | 22 | 19 | 35 | 9 | 50 | 68 | 35 | 11 | 12 | | | |

| PLAYER_NAME | MIN | PTS | REB | AST | FG | FG_PCT | FG3 | FG3_PCT | FT | FT_PCT | STL | TO | BLK | OREB | DREB | PF | START_POSITION | TEAM_CITY |
|---|---|---|---|---|---|---|---|---|---|---|---|---|---|---|---|---|---|---|
| **Jazz** | | | | | | | | | | | | | | | | | | |
| Enes Kanter | 37 | 25 | 8 | 0 | 10-17 | 59% | 1-3 | 33% | 4-4 | 100% | 0 | 1 | 0 | 0 | 8 | 2 | C | Utah |
| Gordon Hayward | 35 | 18 | 8 | 1 | 6-13 | 46% | 2-7 | 29% | 4-4 | 100% | 0 | 3 | 0 | 0 | 8 | 1 | F | Utah |
| Alec Burks | 32 | 16 | 4 | 3 | 4-10 | 40% | 2-6 | 33% | 6-8 | 75% | 1 | 0 | 0 | 1 | 3 | 4 | G | Utah |
| Rudy Gobert | 26 | 9 | 11 | 4 | 2-2 | 100% | 0-0 | 0% | 5-6 | 83% | 1 | 0 | 5 | 4 | 7 | 2 | N/A | Utah |
| Trey Burke | 33 | 7 | 2 | 4 | 3-12 | 25% | 1-5 | 20% | 0-0 | 0% | 1 | 4 | 0 | 1 | 1 | 1 | G | Utah |
| Dante Exum | 15 | 5 | 0 | 2 | 2-4 | 50% | 1-3 | 33% | 0-0 | 0% | 0 | 0 | 0 | 0 | 0 | 0 | N/A | Utah |
| Rodney Hood | 19 | 5 | 1 | 0 | 2-6 | 33% | 1-3 | 33% | 0-0 | 0% | 0 | 1 | 0 | 0 | 1 | 5 | N/A | Utah |
| Derrick Favors | 10 | 4 | 2 | 0 | 2-5 | 40% | 0-0 | 0% | 0-0 | 0% | 0 | 0 | 1 | 2 | 0 | 1 | F | Utah |
| Trevor Booker | 14 | 4 | 4 | 0 | 2-5 | 40% | 0-0 | 0% | 0-0 | 0% | 0 | 1 | 0 | 2 | 2 | 1 | N/A | Utah |
| Jeremy Evans | 6 | 2 | 4 | 0 | 1-2 | 50% | 0-0 | 0% | 0-0 | 0% | 0 | 1 | 0 | 2 | 2 | 1 | N/A | Utah |
| Joe Ingles | 11 | 0 | 0 | 2 | 0-0 | 0% | 0-0 | 0% | 0-0 | 0% | 1 | 0 | 0 | 0 | 0 | 0 | N/A | Utah |
| Steve Novak | 3 | 0 | 0 | 0 | 0-1 | 0% | 0-1 | 0% | 0-0 | 0% | 0 | 0 | 0 | 0 | 0 | 0 | N/A | Utah |
| Ian Clark | N/A | N/A | N/A | N/A | N/A | N/A | N/A | N/A | N/A | N/A | N/A | N/A | N/A | N/A | N/A | N/A | N/A | Utah |
| **Heat** | | | | | | | | | | | | | | | | | | |
| Dwyane Wade | 35 | 29 | 3 | 7 | 10-16 | 63% | 1-1 | 100% | 8-12 | 67% | 1 | 3 | 1 | 0 | 3 | 1 | G | Miami |
| Chris Bosh | 36 | 22 | 9 | 1 | 9-16 | 56% | 1-5 | 20% | 3-4 | 75% | 1 | 1 | 2 | 0 | 9 | 4 | C | Miami |
| Luol Deng | 34 | 14 | 6 | 2 | 5-7 | 71% | 1-2 | 50% | 3-4 | 75% | 1 | 2 | 0 | 2 | 4 | 1 | F | Miami |
| Justin Hamilton | 29 | 9 | 4 | 1 | 4-10 | 40% | 0-2 | 0% | 1-2 | 50% | 1 | 1 | 0 | 1 | 3 | 5 | F | Miami |
| Norris Cole | 28 | 9 | 1 | 3 | 3-6 | 50% | 1-2 | 50% | 2-2 | 100% | 1 | 0 | 0 | 0 | 1 | 1 | G | Miami |
| Danny Granger | 14 | 6 | 2 | 0 | 2-4 | 50% | 2-3 | 67% | 0-0 | 0% | 0 | 0 | 0 | 2 | 0 | 0 | N/A | Miami |
| Udonis Haslem | 17 | 5 | 3 | 0 | 1-4 | 25% | 0-0 | 0% | 3-5 | 60% | 1 | 0 | 0 | 2 | 1 | 1 | N/A | Miami |
| Mario Chalmers | 24 | 4 | 3 | 3 | 1-7 | 14% | 1-4 | 25% | 1-2 | 50% | 0 | 2 | 0 | 0 | 3 | 1 | N/A | Miami |
| Shawne Williams | 13 | 2 | 3 | 1 | 1-1 | 100% | 0-0 | 0% | 0-0 | 0% | 0 | 0 | 0 | 0 | 3 | 2 | N/A | Miami |
| Shabazz Napier | 10 | 0 | 1 | 1 | 0-1 | 0% | 0-1 | 0% | 0-0 | 0% | 0 | 0 | 0 | 0 | 1 | 0 | N/A | Miami |
| Josh McRoberts | N/A | N/A | N/A | N/A | N/A | N/A | N/A | N/A | N/A | N/A | N/A | N/A | N/A | N/A | N/A | N/A | N/A | Miami |
| James Ennis III | N/A | N/A | N/A | N/A | N/A | N/A | N/A | N/A | N/A | N/A | N/A | N/A | N/A | N/A | N/A | N/A | N/A | Miami |
| Hassan Whiteside | N/A | N/A | N/A | N/A | N/A | N/A | N/A | N/A | N/A | N/A | N/A | N/A | N/A | N/A | N/A | N/A | N/A | Miami |

Analyze the box-score and line-score tables for the NBA game between Cavaliers and 76ers on 20th December, 2015 and use the statistics to create a detailed summary of this game.

The Miami Heat (12-11) defeated the Utah Jazz (10-12) 105-95 on Friday. The Heat were able to pull away in the fourth quarter, outscoring the Jazz 31-19. The Jazz were able to keep the game close until the fourth quarter, but they were unable to keep up with the Heat. Miami has won three straight games and is now 12-11 on the season. The Heat will look to keep the win streak going on Saturday when they take on the New York Knicks. Utah has lost two straight games and will look to get back on track on Saturday when they take on the Los Angeles Clippers.**(BLEU=3.32,9-12points)**

The Miami Heat (10–12) defeated the Utah Jazz (6–16) 101–91 on Friday. Miami has won two straight games. They've been playing without Chris Bosh (calf) and Dwyane Wade (hamstring) for the last two games, but they've been able to pull out wins over the Los Angeles Clippers and the Jazz. Mario Chalmers has been filling in admirably at point guard, putting up 20 points and 10 assists in 36 minutes Friday. The Heat will host the New York Knicks on Sunday. Utah has lost four straight games. They've been playing without Derrick Favors (back) for the last three games, and they've lost all three games. Trey Burke has been starting at point guard in Favors' absence, but he's struggled, shooting 10–for–34 from the field and 1–for–10 from beyond the arc over the last three games. The Jazz will host the Phoenix Suns on Saturday.**(BLEU=3.84,7-11points)**

Figure 13: A case study from the Rotowire dataset.

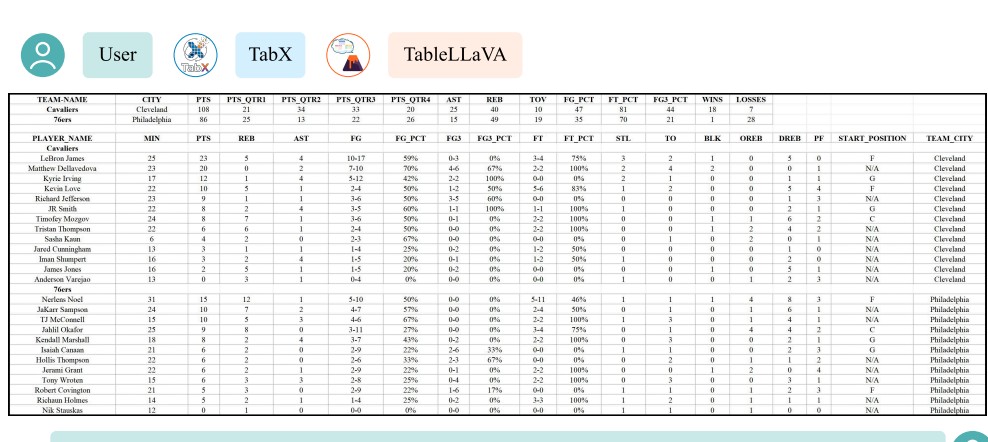

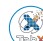

Analyze the box-score and line-score tables for the NBA game between Cavaliers and 76ers on 20th December, 2015 and use the statistics to create a detailed summary of this game.

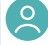

The Cleveland Cavaliers (18-6) defeated the Philadelphia 76ers (5-23) 111-93 on Friday. Cleveland has won 10 of its last 11 games and has now won 17 of its last 18 home games. The Cavaliers were led by LeBron James, who scored 23 points (9-17 FG, 1-2 3Pt, 4-4 FT) and added 10 rebounds and five assists in 36 minutes. Kevin Love added 20 points (8-14 FG, 2-3 3Pt, 2-2 FT) and 10 rebounds in 31 minutes. The Cavaliers will look to continue their winning ways on Sunday against the Boston Celtics. Philadelphia has lost 10 of its last 11 games and has now lost 12 of its last 14 games. The Sixers were led by Nerlens Noel, who scored 18 points (8-14 FG, 0-2 3Pt, 2-2 FT) and added 10 rebounds and two blocks in 31 minutes. Robert Covington added 17 points (7-14 FG, 2-3 3Pt, 0-0 FT) and six rebounds in 33 minutes. The Sixers will look to end their losing streak on Saturday against the New York Knicks.**(BLEU=4.81,20-41points)**

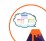

The Cleveland Cavaliers (19-5) defeated the Philadelphia 76ers (1-26) 116–95 on Sunday. The Cavs were expected to win this game easily and they did just that. They led by 10 after the first quarter and never looked back, as they won all four quarters by at least five points. The Cavs shot 52 percent from the field and 50 percent from three-point range. They also dominated the rebounding, as they snatched down 45 boards to the Sixers 31. The Cavs have now won 11 of their last 12 games. LeBron James was the player of the game, as he tallied 26 points, 10 rebounds and six assists in the win. Kyrie Irving also played well, as he amassed 22 points, four rebounds and four assists. Kevin Love recorded a double-double, as he totaled 15 points and 10 rebounds. The Cavs will look to keep rolling on Tuesday against the New York Knicks. The Sixers have been playing better recently, but this was a tough task against the best team in the NBA. Jahlil Okafor led the team in scoring, as he dropped 19 points and 10 rebounds in the loss. Ish Smith was the only other starter in double figures, as he accumulated 14 points, four rebounds and six assists. Nerlens Noel was the only other player in double figures, as he amassed 11 points and 10 rebounds off the bench. Philadelphia will look to play better on Tuesday against the New York Knicks.**(BLEU=6.71,12-32points)**

Figure 14: A case study from the Rotowire dataset.

