# OpenReview forum: "TabX: X-cellent at Complex Tables and Beyond"
_ICLR.cc/2026/Conference — ICLR 2026 Conference Withdrawn Submission_

### Official Review · Reviewer_hFAt · 2025-10-28

**Soundness:** 2
**Presentation:** 3
**Contribution:** 1
**Rating:** 2
**Confidence:** 3

**Summary:**

This paper proposes a solution encompassing high-quality dataset construction and a model fine-tuning framework. It addresses the limitations of existing multimodal large language models (MLLMs) for table understanding, specifically their insufficient robustness to complex table layouts and poor generalization to unseen tasks.

**Strengths:**

- This paper directly targets the core pain points of existing table MLLMs—low-quality training data and inadequate table image understanding.
- The two-stage reflection enhancement strategy of the added tasks achieves quantitative optimization of data quality, integrating model feedback into the data optimization process.

**Weaknesses:**

There are two core concerns which from my respective are very important issues. Solving these concerns may have a great influence of my rating. Concerns are as follows:
- It seems that there are some problems with experimental results. For instance, in Table 1, the performance on the WTQ dataset stands in striking contrast to findings from prior works. While existing studies typically report an average accuracy exceeding 70%—and in some cases reaching 80%—for this work, the accuracy achieved in the present work is merely 18.6%. Same problems exists in some other datasets. Moreover, the performance of TabX is not very good compared to other existing methods. I wonder how authors will explain this concern.
- This paper claims that the pain points of table MLLMs are low-quality training data and inadequate table image understanding. "Inadequate table image understading" is vague, it's better to claim which aspect (semantic or structual). It seems that the method proposed in this paper only solve the low-quality training data. I wonder how the methods solve the problem of inadequate table image understanding.

**Questions:**

Please refer to section Weaknesses.

---

### Official Review · Reviewer_jdan · 2025-10-28

**Soundness:** 3
**Presentation:** 1
**Contribution:** 2
**Rating:** 2
**Confidence:** 4

**Summary:**

This paper presents a multi-modal LLM method similar to TableLlava model on general table understanding. The authors identify two issues in the existing TableLlava model, including the quality in the training dataset, and lack of understanding for complicated table structures. The authors first complement the original MMTab dataset by introducing new tasks (header value matching, data imputation, and NL2SQL). After fine-tuning, the authors employ self evolution with teacher tuning framework to fine-tune the model, where the authors first leverage the student (the trained) model to produce an output and decide if it is a good or bad output. When the example is identified as a bad case, the framework introduces the teacher model to correct the output. The model is tuned in such a refined fashion.

**Strengths:**

- The paper identifies some issues in an existing multimodal table training dataset (MMTab).
- The paper proposes to augment the existing dataset by proposing new tasks (in multi-modal table understanding setup).
- The paper introduces a training framework where refinement is involved.

**Weaknesses:**

### Content

- The fundamental issue with this paper lies in the novelty. Instead of some creative idea in table modeling, the paper seems to be rather an ensemble of data engineering, training engineering. For instance, the three tasks introduced to augment the MMTab dataset, HeaderValueMatching, DataImputation, and NL2SQL, exist in the Table-GPT [1] paper from Microsoft. Though applied in multi-modal setup, I encourage the authors to consider a wider range of tasks discussed in [1]. The authors mentioned that the base model selection is important, which is not a new finding in table modeling aspect [2]. In terms of the feedback loop, [3] provides similar method on progressive data synthesis. In terms of the complicated multi-model table understanding, [4] crafted a nice benchmark which is not discussed or mentioned in this paper. Also, [5] is well worth mentioned in multi-modal table understanding (comparing vision versus text). There are also advances in table modeling both in terms of benchmarking and table LLMs that are worth mentioning.

- The performance improvement seems limited. Of course, kudos to the authors for including such a comprehensive evaluation on a wide range of benchmarks. However, as we understand, most of the applications focus on TQA, TFV, T2T (academic tasks in Table 1), but it seems that the improvement on WTQ (18.6 v.s. Table-LLaVA's 18.4) is limited. And for cases in Table Fact Verfication (TFV), it is Table-LLaVA that takes the lead in most cases). I wonder the effort spent in data and training justify the improvement that we see in Table 1.

- No ablation studies. In terms of all of the methods employed, the authors fail to provide a clear picture into what methods contribute to the improvement. For instance, can the introduction of DataImputation be the source of hallucination rather than the factor for performance improvement? Since in Table 1, no text-to-SQL tasks are involved, how does the introduction of text-to-SQL serve as the source of improvement? Failing to disentangle the core factors for performance improvement makes this paper less informative.


----
### Writing

- Please fix the citations (e.g. Zheng et al. Zheng et al. (2024) at line 050).
- In Table 1, HiT, Table-LLaMA + OCR achieves the best overall score, but the Table-LLaVA's performance is the one that is highlighted.
- Please improve the presentation of the paper in general.


----
### References
[1] Li, Peng, et al. "Table-gpt: Table fine-tuned gpt for diverse table tasks." Proceedings of the ACM on Management of Data 2.3 (2024): 1-28.

[2] Deng, Naihao, et al. "Towards Better Understanding Table Instruction Tuning: Decoupling the Effects from Data versus Models." arXiv preprint arXiv:2501.14717 (2025).

[3] Zheng, Mingyu, et al. "TableDreamer: Progressive and Weakness-guided Data Synthesis from Scratch for Table Instruction Tuning." arXiv preprint arXiv:2506.08646 (2025).

[4] Titiya, Prasham Yatinkumar, et al. "MMTBENCH: A Unified Benchmark for Complex Multimodal Table Reasoning." arXiv preprint arXiv:2505.21771 (2025).

[5] Deng, Naihao, et al. "Tables as texts or images: Evaluating the table reasoning ability of llms and mllms." arXiv preprint arXiv:2402.12424 (2024).

**Questions:**

Other than the questions raised in the weakness parts,

- Why using BLEU scores for the text-to-text generation tasks?
- Does DataImputation lead to hallucinations for the model?

---

### Official Review · Reviewer_HXbJ · 2025-10-30

**Soundness:** 2
**Presentation:** 3
**Contribution:** 2
**Rating:** 2
**Confidence:** 5

**Summary:**

This paper introduces TabX, a MLLM for table understanding. The authors identify two major bottlenecks in existing table MLLMs:
	(1) low-quality instruction–table–answer triplets, and
	(2) insufficient understanding of table images.
To address these challenges, they propose a two-part framework:
Self-Evolution with Teacher-Tuning (SETT) -- a fine-tuning process where the student model performs self-evaluation to identify bad cases, and a teacher model revises these cases to improve data quality and model robustness.
Expanded Instruction-Tuning Dataset (MMTab-Pro) -- constructed by introducing three additional challenging table understanding tasks: HeaderValueMatching, DataImputation, and NL2SQL.
Extensive experiments on the MMTab-eval benchmark demonstrate that TabX consistently outperforms prior table understanding models, showing strong robustness and generalization, particularly on complex and unseen tasks.

**Strengths:**

(1) New methodological integration: Combines teacher–student reflection with iterative self-evolution, which is rarely explored in multimodal contexts.

(2) Strong empirical validation: Comprehensive experiments across 17 tasks, held-in/out settings, robustness and generalization tests. Huge performance improvement.

(3) High-quality dataset contribution: MMTab-Pro meaningfully improves existing instruction datasets both in scale and data clarity.

**Weaknesses:**

(1)	The comparison set is outdated. Table-LLaVA is from early 2024 and no longer represent the current state of multimodal large language models.

For a fairer comparison, you may consider referencing or benchmarking against recent multimodal LLMs such as Qwen3-VL, InternVL2.5, or works like (1) HIPPO: Enhancing the Table Understanding Capability of Large Language Models through Hybrid-Modal Preference Optimization(arXiv:2502.17315) (2) Multimodal Tabular Reasoning with Privileged Structured Information (arXiv:2506.04088)
In addition, including results from strong closed-source MLLMs such as Gemini 2.5 Pro, GPT-4o, or SEED would provide a more comprehensive view of relative performance.

(2)	The SETT framework heavily relies on a strong teacher model. If the teacher is weak or biased, its errors or stylistic preferences may propagate to the student, even after IFD filtering. The observed performance drop in later iterations in Figure7 may reflect this issue.
The reflection-based data enhancement may primarily optimize samples to better fit the student model’s IFD metric, rather than genuinely improving their objective quality. As a result, it is unclear whether the enhanced data are truly stronger or merely more compatible with the model’s existing biases, making the underlying mechanism conceptually ambiguous.

(3)	The paper introduces three new fine-tuning tasks but provides limited discussion on why these particular tasks were chosen. A clearer explanation of the task selection rationale and how they relate to the overall table understanding objectives would strengthen the work.

(4)	The paper mentions total training cost (~$200), but lacks analysis on training time, iteration efficiency, or scalability beyond 7B models.

**Questions:**

(1) The SETT framework updates both the instruction and answer during the self-evaluation phase, but only the answer during the teacher-forced revision phase. Moreover, across iterations, the process alternates between revising instructions and answers.
Could the authors clarify why this specific update order was chosen?
Have alternative update strategies (e.g., revising both jointly or reversing the order) been explored?

(2) How sensitive is the SETT framework to the choice of teacher model? Would smaller or weaker teachers other than Qwen-VL-Max still yield measurable improvement?

(3) Given that the experiments were conducted on 4×A800 GPUs, could the authors comment on the computational efficiency of the SETT iterations? While the total reported cost is $200, the iterative self-reflection–retraining pipeline seems potentially expensive in practice.

---

### Official Review · Reviewer_xQgs · 2025-10-30

**Soundness:** 2
**Presentation:** 3
**Contribution:** 2
**Rating:** 4
**Confidence:** 4

**Summary:**

This paper introduces TabX, a multimodal table understanding model designed to address the limited generalization ability of existing multimodal large language models in handling complex table structures and unseen tasks. Optimizations have been made across data, task definitions, and training methods.
The visual modality of tables represents an important subfield in table-related research, as tabular images can tackle certain challenges that text-based tables struggle with.
While the paper is well-structured, the experimental work is somewhat insufficient, and the analysis lacks depth.

**Strengths:**

1. **Accurate Problem Identification**
The paper precisely identifies key limitations in previous multimodal table models and proposes targeted solutions to address them.

2. **Knowledge Distillation and Self-Evolution**
Despite using a model with relatively few parameters, the proposed approach achieves strong performance by leveraging knowledge distillation from teacher models and a self-evolution algorithm.

3. **Effective Visualizations**
The paper includes a variety of clear and informative figures and tables that enhance the understanding of the methodology and results.

**Weaknesses:**

1. **Limited Table Rendering Methods**
    The study relies solely on table rendering based on the Table-GPT dataset, with optimizations tailored exclusively to this single rendering approach.

2. **Insufficient Model Comparisons**
    The experimental section includes comparisons with only a limited number of baseline models.

3. **Lack of Justification for Teacher Model Selection**
    The rationale for selecting Qwen-VL-Max as the teacher model is not clearly explained, and its performance on the relevant benchmarks remains unstated.

4. **Superficial Experimental Analysis**
    The analysis lacks depth in several aspects—for example, it does not thoroughly investigate the performance drop during multi-round iterations, and the evaluation of TabX's performance on RW is weakly supported. Incorporating human evaluation or other detailed analytical methods could help better interpret the results.

**Questions:**

1. **Prompt Format Specification for TableLLaMA**
    The performance of TableLLaMA is known to be highly sensitive to prompt formatting. Could the authors clarify whether the official recommended prompts were used in their experiments? If non-standard prompts were adopted, it would be important to explain how this may have influenced the results.

2. **Scope of Baselines and OCR Techniques**
    As the experiments were likely conducted before some recent advances, the study does not include several newer models—particularly other multimodal methods for table understanding. Moreover, the OCR approach used is not described in detail. Many contemporary OCR systems extend beyond plain text recognition and support structured output such as LaTeX. We suggest the following additional comparisons:
    - TableLLaMA enhanced with modern OCR,
    - TableLLaMA with direct text-table input, and
    - More comprehensive baselines covering both text-based and multimodal table models.
    These comparisons are critical because, in the absence of (1) and (2), it remains unclear whether the reported superiority of the proposed method holds when TableLLaMA is provided with its optimal input (i.e., plain text tables). It is plausible that TableLLaMA could outperform the TabX under such conditions.

3. **Selection of the Base Model**
    Given that Janus-Pro underperforms relative to Table-LLaVA, could the authors explain the rationale for choosing Janus-Pro as the base model?

4. **Table Rendering Strategy**
    The rendering method used for tabular images may affect model performance. Have the authors considered whether some baseline MLLMs performed poorly due to suboptimal rendering? Additional experiments analyzing different rendering strategies would strengthen the study.

---

### Note · Authors · 2026-01-27

I have read and agree with the venue's withdrawal policy on behalf of myself and my co-authors.

---

### Meta-Review · Area_Chair_8qCm · 2025-12-23

**Summary:**

The paper "TabX: X-cellent at Complex Tables and Beyond" introduces a multimodal large language model designed to enhance table understanding by addressing limitations in existing methods through a two-step pipeline: first, curating the MMTab-Pro dataset with reflective enhancement and three new tasks (HeaderValueMatching, DataImputation, and NL2SQL) to improve data quality, and second, proposing a Self-Evolution with Teacher-Tuning (SETT) framework that enables iterative model refinement via self-evaluation and teacher guidance, with experiments showing superior performance on complex and unseen tasks in the MMTab-eval benchmark. Reviewer opinions are divided, highlighting strengths such as accurate problem identification, strong empirical results, and valuable dataset contributions, but also criticizing weaknesses including limited novelty due to similarities with prior work, outdated baseline comparisons, lack of ablation studies, and vague technical justifications, resulting in ratings ranging from rejection to marginal acceptance; however, no formal rebuttal is provided by the authors, though potential responses could focus on clarifying innovative aspects of SETT, expanding experimental scope with state-of-the-art models, and addressing analytical gaps to strengthen the paper's contribution. Based on the reviewers' comments, the paper is not yet ready for acceptance.

**Reviewer Concerns:**

No formal rebuttal is provided.

**Reviewer Scores:**

No formal rebuttal is provided.

---

### Decision · Program_Chairs · 2026-01-26

Reject